# The effectiveness of mindfulness training in improving medical students' stress, depression, and anxiety

**Ahmed M. Alzahrani**[1,2]*, **Ahmed Hakami**[3], **Ahmad AlHadi**[4,5], **Nassr Al-maflehi**[6], **Mohammed H. Aljawadi**[7], **Rawan M. Alotaibi**[8], **Muhannad M. Alzahrani**[8], **Salem Ali Alammari**[8], **Mohammed A. Batais**[1,2], **Turky H. Almigbal**[1,2]

**1** University Family Medicine Center, King Saud University Medical City, King Saud University, Riyadh, Saudi Arabia, **2** Department of Family and Community Medicine, College of Medicine, King Saud University, Riyadh, Saudi Arabia, **3** Riyadh Regional Laboratory, Ministry of Health, Riyadh, Saudi Arabia, **4** Department of Psychiatry, College of Medicine, King Saud University, Riyadh, Saudi Arabia, **5** SABIC Psychological Health Research & Applications Chair, College of Medicine, King Saud University, Riyadh, Saudi Arabia, **6** Department of Periodontics and Community Dentistry, College of Dentistry, King Saud University, Riyadh, Saudi Arabia, **7** Department of Clinical Pharmacy, College of Pharmacy, King Saud University, Riyadh, Saudi Arabia, **8** College of Medicine, King Saud University, Riyadh, Saudi Arabia

* a429mz@gmail.com

**Data Availability Statement:** All relevant data are within the paper and its Supporting Information files.

## Abstract

### Introduction

There is growing interest in Mindfulness-based Stress Reduction (MBSR) program to combat mental distress in medical students. In Saudi Arabia, literature is insufficient about MBSR and its effectiveness. This study aims to measure the effectiveness of MBSR in improving mindful state, stress, anxiety, and depression in medical students. Also, the study explores the association between the attendance rate of MBSR sessions and its effectiveness. Lastly, the study examines gender differences in response to MBSR.

### Methods

This is a stratified randomized controlled study of 84 medical students from two medical schools in Riyadh, Saudi Arabia. They were recruited voluntarily from November 2018 to April 2021, and allocated to MBSR and waitlist groups using a stratified randomization method based on gender. MBSR group received eight weeks of sessions through audiovisual materials. An online survey utilizing validated questionnaires assessing stress, mindfulness, anxiety, and depression was used to evaluate both groups pre-program (time 0), post-program (time 1), and three months later (time 2).

### Results

Seventy-one participants completed the post-test (time 1). There were no differences between study groups at time 0 and 1. However, in 41 subjects who completed the follow-up test (time 2), the anxiety dropped significantly in MBSR group (mean difference (MD), -3.935; 95% CI, -7.580 to -0.290). Furthermore, attending more MBSR sessions was

**Funding:** This research was funded by Dallah Healthcare Grant. project number: CMRC-DHG-1/ 002. The funders had no role in study design, data collection and analysis, decision to publish, or preparation of the manuscript.

**Competing interests:** The authors have declared that no competing interests exist.

inversely correlated with depression (r, -0.556; P, 0.002), and anxiety (r, -0.630; P, 0.000). Compared to their baseline, males in MBSR group improved in stress (MD, 3.08; 95% CI, 0.30 to 5.86), anxiety (MD, 4.91; 95% CI, 3.32 to 6.50), and mindfulness (MD, -0.58; 95% CI, -1.01 to -0.15), while females improved in stress (MD, 2.64; 95% CI, 0.02 to 5.26).

## Conclusion

Despite the study being interrupted by the stressful COVID-19 outbreak, the findings suggest that MBSR improved psychological outcomes when participants commit to the program.

## Introduction

Stress, anxiety, and depressive symptoms are prevalent conditions in medical education and have been investigated thoroughly in the literature [1–5]. Regionally, Abdulghani et al. found that the prevalence of stress among medical students was 63% and significantly higher in females [6]. On the other hand, AlFaris et al. found a high prevalence of depression among students of dentistry (51.6%), medicine (46.2%), applied medical sciences (45.7%), and nursing (44.2%), and was significantly higher among females [7]. Another local study screened medical students for anxiety and depression and found significant gender differences in the prevalence of anxiety and depression (females 66.6%, males 44.4%), and the rate was much higher in the first year (females 89.7%, males 60%) [8].

These alarming findings, if unattended, are associated with job-related burnout syndrome [9]. As a consequence, this could affect work performance [10, 11], reduce job dedication [12], decrease self-satisfaction [13], delay recovery for patients after discharge from the hospital, reduce patient satisfaction [14], decrease productivity, and increase medical errors [15]. The seeds of this issue could be implanted early during undergraduate years, which require a need-driven response with convenient solutions. Mindfulness is an example of such a solution and can be defined as a mental process or form of mental exercise with the aim of enhancing the state of being attentive to and aware of the present, as well as disengaging from non-adaptive thoughts and emotions that make one vulnerable to stress [16, 17]. Multiple studies have revealed an inverse relationship between mindfulness and stress, depression, and anxiety in different populations [18–20]. Locally, two published studies targeting medical students and residents have highlighted similar findings, that mindfulness was inversely correlated with stress and depressive symptoms [21, 22].

Kabat-Zinn at the University of Massachusetts Medical Center has developed a Mindfulness-Based Stress Reduction (MBSR) program as an intervention [23]. This program was designed to teach participants to become more aware of and relate differently to thoughts, feelings, and all the stimuli that enter their perimeter of awareness, as well as to see their habitual reactions to stress. Thus, it works as an antidote for maladaptive patterns of thinking such as rumination, which is believed to be one of the sources of psychological stress and is associated with depression and anxiety [24, 25].

The utilization of mindfulness-based interventions has been shown to be beneficial in different populations. In Spain [26], a study of 68 primary care physicians (43 in the intervention group and 25 in the control group), concluded that the intervention group improved in the 4 scales measured. The magnitude of the change was large in total mood disturbance and mindfulness, while it was moderate in the burnout and empathy scales. Another study that involved

66 medical students at the University of Tasmania, has utilized an exclusive audio-guided mindfulness practice as an intervention. The study findings that the intervention group was improved in the stress and anxiety, and it was found to be maintained after 8 weeks [27].

In another study, 288 medical and psychology students from the University of Oslo and the University of Tromso were randomly allocated to an MBSR group or a control group. Subjects were evaluated using multiple questionnaires including perceived medical school stress and five-facet mindfulness. Among female students, a significant improvement was noted in the mindfulness facets of non-reacting, non-judging, and lowered stress. Better adherence to the mindfulness program and exercises was thought to be a predictor of such improvement [28].

Discussion of mindfulness training in negation of detrimental effect of mental distress in literature has increased our scientific curiosity about its effectiveness in Saudi medical colleges. To our knowledge, publications from Saudi Arabia addressing the effect of MBSR program on medical students are still in the early stages. The purpose of current study was to examine the effectiveness of the MBSR program in improving mindful state, stress, anxiety, and depression in medical students at Vision Colleges and King Saud University (KSU). Also, the study aimed to assess whether the effect of the MBSR program was maintained three months after the end of the program. Additionally, the study aimed to explore the association between the attendance rate of MBSR sessions and its effectiveness on all study measures. Lastly, it aimed to examine the gender differences in response to the program in all study's measures.

## Methods

### Study design and procedure

This stratified randomized study was designed with pre-test (time 0), post-test (time 1), and follow-up measurements (time 2). The target population was medical students at KSU, which is a governmental medical school, and Vision private colleges in Riyadh, Saudi Arabia. The college of medicine at King Saud University offers medical education over a period of 6 years where the first two years tackle basic medical sciences and the next 3 years reserved for clinical education and bedside teaching. The curriculum relies greatly on lectures, problem-solving sessions, and bedside teaching. Passing the last year of internship training is required before receiving the Bachelor of Medicine, Bachelor of Surgery (MBBS). There was a shift to online education during the COVID-19 pandemic. The medical college in Vision private colleges share similar education system. A sample size calculation concluded that at least 37 participants per group would be required to detect a mean difference of 2 in our primary outcome which was the perceived stress scale (PSS), with a power of 90% at a significance level of 0.05. The sample size was increased to 42 subjects per group to compensate for possible dropout. Participants were recruited between November 2018 and April 2021 on a voluntary basis. We invited the students to participate through advertising lectures and brochures, after the IRB approval for the content of that lecture and the brochure. We announced the lecture through the student council. It was conducted by one of the research team members, who is a consultant and associate professor in psychiatry. Before the COVID-19 outbreak, it was an on-site lecture at King Saud University Medical School and Vision Colleges. Then the lecture was repeated online during the pandemic to recruit more candidates. The lecture aimed to increase the awareness of stress and its impact on the student's well-being and offered evidence-based solutions to cope with it. Then it elaborated more on the concept of mindfulness, the aim of the study, and inclusion/exclusion criteria. The lecture ended up with a QR code to guide the students who are interested in joining the study to the invitation link to the study. The link collected the contact information of the candidates. Those who filled out the link were contacted, and if they fit the study's criteria, we invited them to sign the consent form and answered their

inquiries about the study. A brochure was also used to invite possible candidates through the student council database of students' emails. In order to minimize the dropout, we discussed participants' expectations before the study started, addressed their concerns, identified possible difficulties that could affect their participation.

The inclusion criteria were current active status as medical student at Vision Colleges or KSU and willingness to participate and commit fully to the study. The exclusion criteria were inability to speak, read, and write in English, having any plans that could interfere greatly with the study adherence (such as surgical procedures or traveling), currently practicing meditation or involvement in any stress reduction program, and receiving any psychiatric treatment.

There were 84 subjects who enrolled in the study after they consented, completed the pre-test measures, stratified randomization based on gender was used to allocate the 84 medical students to the study groups. The stratified randomization process was concealed and performed by an external party, two randomly generated lists using computer software were used to allocate males and females participants separately. There were 39 participants allocated to the MBSR group and 45 in the waitlist group. Ideally, the stratified randomization works better when all the subjects are available and enrolled at the same time. But in our case, the COVID-19 outbreak impacted the recruitment and we had to start with the consented participants and continue working to recruit more. We believed that we might lose our participants if we kept them waiting till we got our targeted sample size. So, the stratified randomization ended up with different numbers allocated in both groups. However, we reached the least required number in each group of the calculated sample size which is 37 participants.

The MBSR group received an 8-week online MBSR program, while the other group received the same program at only the end of the study as a means of gratitude for their participation. The wait-list group were involved in the study to overcome the maturation effect and to maintain the internal validity of the study, ensuring that any improvement in the study outcomes can be attributed to the intervention. Participants were advised not to tell others about the group to which they were allocated and not to share the program contents with others.

The intervention was an 8-week self-paced online MBSR program with 2.5 hours of sessions per week through audiovisual materials. The following section will discuss the intervention in detail. In session one, the trainees are introduced to theory and evidence behind mindfulness, the difference between formal and informal exercises, learning proper posture for exercises, mindful eating, mindful breathing, and body scan. In session two, the trainees start exploring their perception about life events, expanding their awareness of breathing, extended body scan, sitting meditation, and how to be present in the moment. In session three, the trainees get introduced to yoga and its practices. In session four, the trainee learned the physiology of stress. In session five, the trainees learned about mindful response and how to apply mindfulness in critical situations. Session six discussed resilience and returning to baseline after experiencing stressful events and revisiting formal and informal exercises. In session seven, the trainees learned how to acquire a broader sense of awareness, being unjudgmental and strengthen daily practices. In session eight, the trainees learned about loving-kindness and strategies to maintain and deepen skills developed throughout the program.

## Measures

The following English-version scales were used. The Mindful Attention Awareness Scale (MAAS) is a 15-item scale that was used for the measurement of mindfulness, it assesses awareness and attention to day-to-day experiences using a scale from 1 to 6, where 1 represents almost always, and 6 represents Almost never. The final score is the mean of the 15 items. A higher score indicates a better level of mindfulness. The scale shown good validity and

reliability [16, 29]. Stress was measured using PSS, a global measure of perceived stress, it is a valid, reliable, and widely use self-reported scale to evaluate the stress, asking about feeling and thoughts in the last 30 days, rating them from 0 to 4, where 0 reflects never and 4 reflects very often. A total score ranges from 0–13 represents low stress, 14 to 26 means moderate stress, and more than 27 to 40 considered high stress. [30]. The Patient Health Questionnaire 9 (PHQ-9) is a 9-item, valid, and reliable self-reported questionnaire that was used to assess depressive symptoms, it scores each symptom from 0 to 3, where 0 (not at all) and 3 (nearly every day). The total score ranges from 0 to 27, where Scores of 5, 10, 15, and 20 represent cut-offs for mild, moderate, moderately severe, and severe depression, respectively. It can be used to screen and diagnose depression. [31]. The 7-item anxiety scale (GAD-7) was utilized for anxiety assessment and is a self-reported, well-known, valid and reliable tool. It has seven items, each of which is scored 0 to 3, where 0 (not at all) and 3 (nearly every day) providing a 0 to 21 severity score. [32]. No permissions were required to use MAAS, PSS, PHQ-9, and GAD-7 in nonprofit academic research.

The questionnaires were provided to the participants online. In order to minimize missing data, the online questionnaires were designed to ensure that all items must be answered before the participant could move to the next page. The attendance rate to the weekly sessions was monitored by self-reporting the number of sessions attended in the post-test questionnaires (time 1), and the participants were reminded weekly through e-mail.

## Outcomes

Demographic data (age, gender, marital status, undergraduate year, and college) and baseline scores of the mentioned measures were acquired first (time 0). Then, the subjects who completed this stage were allocated to MBSR and waitlist groups. At the end of the program (time 1), the same measures were collected again along with the number of attended sessions. The primary outcome was the difference in means between the post-test results of both groups in the stress scale. The secondary outcomes of the study included the difference in means between the post-test results of both groups in depression, anxiety, and mindfulness status scales. Also, follow-up data which were collected after 3 months (time 2) to observe the ongoing effect of the MBSR program between the study groups, by comparing the means of all study measures. In addition, assessing the association between the attendance rate of the MBSR sessions (from 0 to 8 sessions) with the mean difference between time 0 and time 1 of all study measures in the MBSR group. Lastly, evaluating the difference in means of all study measures among males (time 1 versus time 0) and females (time 1 versus time 0) allocated to the MBSR group.

## Data analysis

Data that were normally distributed were presented as the mean ± standard deviation (SD) for continuous variables and as frequencies and percentages for categorical variables. The Shapiro–Wilk test and the Kolmogorov–Smirnov test were used to evaluate the normality of the data distribution. Comparisons of groups at the three time points were performed using a t-test. Fisher's exact test was used to compare the categorical variables in baseline characteristics of the participants. Spearman's correlation coefficient was used to test the relationship between the attendance to the MBSR program and the change in means of all measures. A paired t-test was utilized to assess the responses among males and females in the MBSR group. One way ANOVA test was utilized to assess the change in means within and between both study groups in time 0, time 1, and time 2. A p-value less than 0.05 was required for statistical significance. Analyses were performed using SPSS software [33].

## Ethical consideration

The study was approved by the institutional review board (IRB) at College of Medicine, KSU (project number: E-17-2419). Written informed consent was obtained on an individual basis. However, an amendment for acquiring informed consent electronically was approved by the IRB office due to the COVID-19 outbreak. Data were electronic in nature and were kept on a private computer, which had no public access and was password protected. The privacy of the research participants was protected, as was the confidentiality of their personal information. The first author (AA) had access to information that identifies individual participants during and after data collection. The consent process and the study data were monitored by the IRB's compliance office.

## Results

The 84 participants' demographic information is presented in Table 1. The majority were female (61.5% in the MBSR group and 66.7% in the waitlist group). The data normality was assumed for all variables except for the attendance rate to the MBSR session. The mean age was 22.15 (SD 1.7) years for the MBSR group and 22.44 (SD 1.7) years for the waitlist group. There were no significant differences between both groups in all variables. The pre-test measures (time 0) are shown in Table 2. There were no significant differences between the study groups in the PHQ-9, MAAS, GAD-7, and PSS scores (p = 0.65, 0.76, 0.65, 0.89, respectively). The study flowchart is shown in Fig 1.

The post-test measures (time 1) of 71 participants are represented in Table 3. The results indicate no significant differences between study groups in PHQ-9, MAAS, GAD-7, and PSS (p = 0.177, 0.054, 0.113, 0.297 respectively). There were 13 participants (15%) who dropped out through nonadherence to post-tests, and 10 of them were from the MBSR group.

There were 41 participants who completed the follow-up measures (time 2), including 16 from the MBSR group. Table 4 shows that the MBSR group had a significantly lower mean GAD-7 score with a difference of -3.935 (p = 0.035). The MBSR group also showed a slightly

**Table 1. Baseline characteristics.**

| Variable | level | Group | | | | Significance |
|---|---|---|---|---|---|---|
| | | MBSR (n = 39) | | Waitlist (n = 45) | | |
| | | Frequency | Percent | Frequency | Percent | F-exact test (p-value) |
| Gender | Male | 15 | 38.5 | 15 | 33.3 | 0.625 |
| | Female | 24 | 61.5 | 30 | 66.7 | |
| Medical college | KSU | 24 | 61.5 | 26 | 57.8 | 0.825 |
| | Vision Colleges | 15 | 38.5 | 19 | 42.2 | |
| Marital status | Married | 1 | 2.6 | 0 | 0.0 | 0.464 |
| | Single | 38 | 97.4 | 45 | 100.0 | |
| Medical year | 1st year | 4 | 10.3 | 2 | 4.4 | 0.678 |
| | 2nd year | 10 | 25.6 | 10 | 22.2 | |
| | 3rd year | 3 | 7.7 | 7 | 15.6 | |
| | 4th year | 13 | 33.3 | 14 | 31.1 | |
| | 5th year | 9 | 23.1 | 12 | 26.7 | |
| Age, years | | Mean | Std. Deviation | Mean | Std. Deviation | Independent T-test (p-value) |
| | | 22.15 | 1.72 | 22.44 | 1.79 | 0.453 |

MBSR: Mindfulness-based stress reduction program, KSU: King Saud University

**Table 2. Pre-test measures (time 0).**

| | Group | | | | Independent T-test | | | |
|---|---|---|---|---|---|---|---|---|
| | MBSR (n = 39) | | Waitlist (n = 45) | | Sig. (2-tailed) | Mean Difference | 95% Confidence Interval of the Difference | |
| | Mean | Std. Deviation | Mean | Std. Deviation | | | Lower | Upper |
| PHQ-9 | 9.95 | 5.81 | 10.51 | 5.55 | 0.65 | -0.56 | -3.03 | 1.90 |
| MAAS | 3.74 | 0.88 | 3.67 | 1.08 | 0.76 | 0.06 | -0.36 | 0.49 |
| GAD-7 | 9.72 | 4.47 | 10.18 | 4.76 | 0.65 | -0.46 | -2.47 | 1.55 |
| PSS | 21.85 | 6.63 | 22.02 | 5.94 | 0.89 | -0.17 | -2.90 | 2.55 |

MBSR: Mindfulness-based stress reduction program, PHQ-9: Patient health questionnaire 9, MAAS: Mindful attention awareness scale, GAD-7: Generalized anxiety disorder scale 7, PSS: perceived stress scale.

lower PSS score with a mean difference of -4.328 (p = 0.053). In contrast, no significant differences were found in PHQ-9 and MAAS scores (p = 0.137, 0.731, respectively).

Table 5 represents the difference within-and between the study subjects at time 0, time 1, and time 2 using one-way ANOVA test and independent t-tests. It showed that there were significant differences within the MBSR group PSS (p = 0.023), and within both groups in GAD-7, MBSR group (p = 0.023) and waitlist group (p = 0.049). Additionally, there was a significant difference between both groups in GAD-7 at time 2 (p = 0.035).

Table 6 shows the number of sessions attended by the participants in the MBSR group. There were 8 participants who attended the whole program, while there were 3 participants who did not attend any sessions. The Spearman correlation coefficient was used to assess the effect of attendance on the change in means of all measures in the MBSR group. Table 7 shows that attending more sessions of the MBSR program was inversely associated with both PHQ-9 (r = -0.556, p = 0.002) and GAD-7 scores (r = -0.630, p = 0.000).

Table 8 presents the differences in means (time 0- time 1) among males and females in the MBSR group according to a paired t-test. Males had significant improvements in MAAS, GAD-7, and PSS scores (p = 0.013, 0.000, and 0.033, respectively), while females had a significant improvement in the PSS score only (p = 0.048).

## Discussion

Our study included subjects from both preclinical (42.8%) and clinical years (57.1%). They were predominantly from the governmental medical college KSU (59.5%) and female (64.2%). The baseline characteristics and the pre-test measures (time 0) indicated no initial difference between the MBSR and waitlist groups. Which is reflecting the desired effect of randomization.

Unexpectedly, the post-test measures (time 1) were not supportive of the study hypothesis. In the MBSR group, three participants did not attend any sessions, and 12 participants attended less than 75% of the sessions. This was expected, given the fact that studying in medical colleges is naturally demanding and connected to long studying hours and limited sleep [1, 34, 35]. Furthermore, the shift to online learning during the COVID-19 pandemic was stressful and created more time-consuming tasks and assignments [36]. In comparison, worldwide, multiple studies have demonstrated the effectiveness of mindfulness training for depressive symptoms [37] and stress [38] among medical students. However, other studies have also reported no significant effect of mindfulness training on depressive symptoms [27] and stress [28].

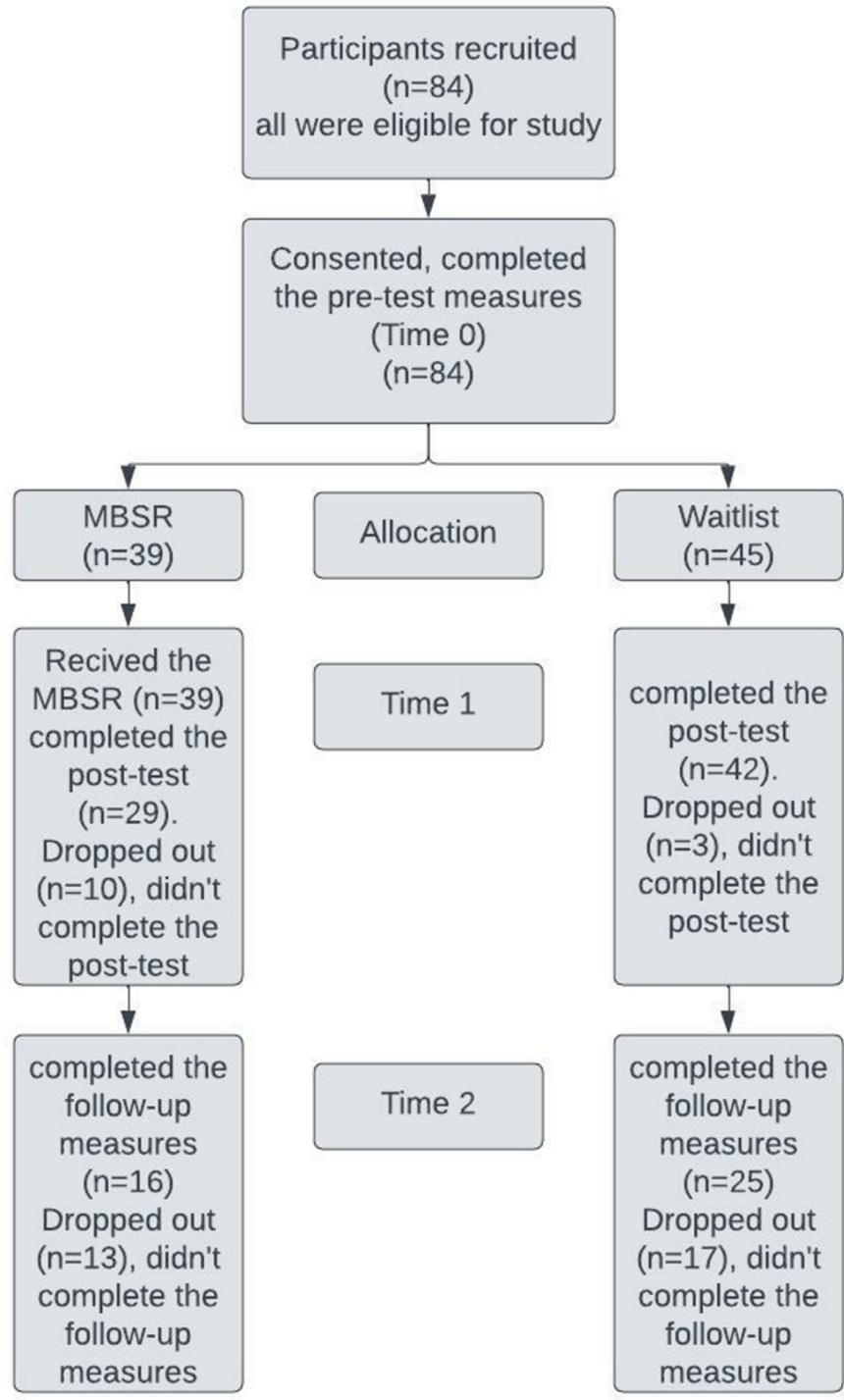

**Fig 1. Flowchart of the study.**

Interestingly, the follow-up measures (time 2) showed better outcomes in terms of mean differences between the groups in depression, anxiety, and stress scales (the mean differences were -2.828, -3.935, and -4.328, respectively). Although these results reflected a favorable direction toward better outcomes, only anxiety had a statistically significant change. Equally

**Table 3. Post-test measures (time 1).**

| | Group | | | | Independent T-test | | | |
| | MBSR (n = 29) | | Waitlist (n = 42) | | Sig. (2-tailed) | Mean Difference | 95% Confidence Interval of the Difference | |
| | Mean | Std. Deviation | Mean | Std. Deviation | | | Lower | Upper |
|---|---|---|---|---|---|---|---|---|
| PHQ-9 | 7.14 | 6.19 | 9.02 | 5.38 | 0.177 | -1.886 | -4.644 | 0.872 |
| MAAS | 4.18 | 0.94 | 3.72 | 0.97 | 0.054 | 0.457 | -0.007 | 0.922 |
| GAD-7 | 6.45 | 4.97 | 8.48 | 5.39 | 0.113 | -2.028 | -4.546 | 0.490 |
| PSS | 18.69 | 7.87 | 20.52 | 6.75 | 0.297 | -1.834 | -5.315 | 1.647 |

BSR: Mindfulness-based stress reduction program, PHQ-9: Patient health questionnaire 9, MAAS: Mindful attention awareness scale, GAD-7: Generalized anxiety disorder scale 7, PSS: perceived stress scale.

important was that attending more sessions was found to have a desirable effect on depression and anxiety. Similarly, De Vube et al. found that the attendance rate in a mindfulness-based training can predict better outcomes in psychology and medical students [28]. The repeated measurement within-between subject showed that better outcomes in anxiety and stress scales, but this was only limited for those who completed the full assessment at the three time intervals, (n = 41).

Notably, the majority of participants in many studies, including our study, were females [28, 38, 39]. This could be explained by the higher levels of stress among them compared to males [6], which may motivate them to volunteer in such studies. Thus, without gender-based stratified randomization, the generalizability of the findings to males is questionable. We employed stratified randomization by gender in order to assess whether gender would interfere with the outcomes. The analysis of gender-related differences of MBSR effects revealed that males had a significant increase in the post-test level of mindfulness. Additionally, they had a significant reduction in anxiety, and stress. On the other hand, females responded significantly only to stress. A local qualitative design study aimed to explore the effect of acceptance and commitment therapy in group of 8 Saudi female participants with depression and anxiety. The study concluded that mindfulness practice relieved their sleep problems and helped them to feel more relaxed [40].

Obviously, the variation in attendance between males and females may have contributed to these findings. As shown in Table 6, 57% of participants who attended more that 75% of the program were males. In contrast, 73% of participants who attended less than 75% of the sessions, including those who did not attend any sessions, were females. Exploring other factors

**Table 4. Follow-up measures (time 2).**

| | Group | | | | Independent t-test | | | |
| | MBSR (n = 16) | | Waitlist (n = 25) | | Sig. (2-tailed) | Mean Difference | 95% Confidence Interval of the Difference | |
| | Mean | Std. Deviation | Mean | Std. Deviation | | | Lower | Upper |
|---|---|---|---|---|---|---|---|---|
| PHQ-9 | 6.81 | 4.53 | 9.64 | 7.39 | 0.137 | -2.828 | -6.598 | 0.943 |
| MAAS | 3.97 | 0.96 | 3.85 | 1.15 | 0.731 | 0.120 | -0.582 | 0.823 |
| GAD-7 | 6.63 | 4.99 | 10.56 | 5.99 | **0.035** | -3.935 | -7.580 | -0.290 |
| PSS | 19.31 | 7.04 | 23.64 | 6.58 | 0.053 | -4.328 | -8.706 | 0.051 |

MBSR: Mindfulness-based stress reduction program, PHQ-9: Patient health questionnaire 9, MAAS: Mindful attention awareness scale, GAD-7: Generalized anxiety disorder scale 7, PSS: perceived stress scale.

**Table 5. The differences within-and between study subjects.**

|  | MBSR (n = 16) | Waitlist (n = 25) | Independent t-test, p-value |
|---|---|---|---|
| PHQ-9, time 0, mean (SD) | 7.81 (4.26) | 9.8 (5.45) | 0.223 |
| PHQ-9, time 1, mean (SD) | 6.06 (4.4) | 9 (5.63) | 0.085 |
| PHQ-9, time 2, mean (SD) | 6.81 (4.53) | 9.64 (7.4) | 0.137 |
| RM, p-value | 0.345 | 0.608 |  |
| PSS, time 0, mean (SD) | 22.25 (4.75) | 21.44 (5.91) | 0.648 |
| PSS, time 1, mean (SD) | 18.88 (6.42) | 20.96 (7.14) | 0.359 |
| PSS, time 2, mean (SD) | 19.31 (7.04) | 23.64 (6.58) | 0.053 |
| RM, p-value | **0.023** | 0.067 |  |
| MAAS, time 0, mean (SD) | 3.81 (0.75) | 3.74 (0.91) | 0.794 |
| MAAS, time 1, mean (SD) | 4 (0.9) | 3.88 (0.89) | 0.701 |
| MAAS, time 2, mean (SD) | 3.97 (0.96) | 3.85 (1.16) | 0.731 |
| RM, p-value | 0.572 | 0.664 |  |
| GAD-7, time 0, mean (SD) | 8.75 (3.04) | 9.56 (4.41) | 0.524 |
| GAD-7, time 1, mean (SD) | 6.06 (4.49) | 8.12 (5.2) | 0.201 |
| GAD-7, time 2, mean (SD) | 6.63 (4.99) | 10.56 (5.99) | **0.035** |
| RM, p-value | **0.023** | **0.049** |  |

MBSR: Mindfulness-based stress reduction, SD: standard deviation, RM: repeated measure ANOVA.

that may influence the outcomes is recommended for future research in order to identify those who could benefit most from such a program.

## Limitations

One limitation of this study was that the recruitment of participants was challenging. First, it was difficult to convince medical students to join a 5-month study. In addition, the recruitment phase was interrupted by the COVID-19 pandemic, which limited our ability to reach and meet with potential participants. The awareness and knowledge about mindfulness in the region pose another challenge for recruiting subjects easily. Clearly the drop-out number in the MBSR group was higher than the waitlist, which increases the likelihood of type 2 error, and could affect the balance of the baseline characteristics between the study groups. In

**Table 6. The attendance in MBSR group stratified by gender.**

| MBSR sessions | Number of participants (n = 29) | Gender | |
|---|---|---|---|
|  |  | Male (n = 12) | Female (n = 17) |
| Didn't attend any sessions | 3 | 0 | 3 |
| One session | 2 | 1 | 1 |
| Two sessions | 6 | 2 | 4 |
| Three sessions | 0 | 0 | 0 |
| Four sessions | 3 | 1 | 2 |
| Five sessions | 1 | 0 | 1 |
| Six sessions | 1 | 1 | 0 |
| Seven sessions | 5 | 2 | 3 |
| Eight sessions | 8 | 5 | 3 |

MBSR: Mindfulness-based stress reduction

**Table 7. Correlation of attendance with the difference in means between pre- and post-measures in MBSR group (time 1—time 0).**

| | | Attendance | PSS | PHQ-9 | MAAS | GAD-7 |
|---|---|---|---|---|---|---|
| | | Correlations | | | | |
| Attendance | Spearman | 1 | -0.303 | -.556 | 0.351 | -.630 |
| | Sig. (2-tailed) | | 0.111 | **0.002** | 0.062 | **0.000** |
| | N | | 29 | 29 | 29 | 29 |

PHQ-9: Patient health questionnaire 9, MAAS: Mindful attention awareness scale, GAD-7: Generalized anxiety disorder scale 7, PSS: perceived stress scale

addition, we didn't assess the reasons behind the low number of participants who completed the whole program sessions in the MBSR group.

## Recommendations

For future local studies, instead of an 8-week program, a short version of the MBSR could be used to improve participant adherence. In addition, we hope to inspire local experts to consider cultural-based MBSR modifications that best suit the population. For instance, local customs and activities for practicing mindfulness could be employed. We also encourage mental health practitioners to contribute to the population's awareness of mindfulness concepts and practices and to consider mindfulness training programs in the curricula of local medical schools.

## Conclusion

The present study indicated that the MBSR program had a promising positive effect on the mental wellbeing of medical students. The differences were insignificant between both groups

**Table 8. Gender differences in means between pre- and post-measures in MBSR group (time 0—time 1).**

| Paired Samples Statistics | | | | Paired Differences | | Sig. (2-tailed) |
|---|---|---|---|---|---|---|
| Gender | Variable | Time | Mean (SD) | Mean dif. (time0-time1) (SD) | 95% Confidence Interval of the Difference | |
| Male (n = 12) | PHQ-9 | Time 0 | 7.33 (5.77) | 3.25 (5.57) | [-0.29, 6.79] | 0.069 |
| | | Time 1 | 4.08 (2.96) | | | |
| | MAAS | Time 0 | 4.27 (0.89) | -0.58 (0.68) | [-1.01, -0.15] | **0.013** |
| | | Time 1 | 4.86 (0.81) | | | |
| | GAD-7 | Time 0 | 8.25 (3.84) | 4.91 (2.50) | [3.32, 6.50] | **0.000** |
| | | Time 1 | 3.33 (3.39) | | | |
| | PSS | Time 0 | 18 (6.33) | 3.08 (4.37) | [0.30, 5.86] | **0.033** |
| | | Time 1 | 14.92 (7.29) | | | |
| Female (n = 17) | PHQ-9 | Time 0 | 11.53 (6.11) | 2.23 (4.49) | [-0.07, 4.54] | 0.057 |
| | | Time 1 | 9.29 (7.00) | | | |
| | MAAS | Time 0 | 3.48 (0.79) | -0.22 (0.78) | [-0.62, 0.18] | 0.261 |
| | | Time 1 | 3.70 (0.72) | | | |
| | GAD-7 | Time 0 | 10.06 (3.50) | 1.41 (3.72) | [-0.50, 3.32] | 0.138 |
| | | Time 1 | 8.65 (4.79) | | | |
| | PSS | Time 0 | 24 (5.78) | 2.64 (5.09) | [0.02, 5.26] | **0.048** |
| | | Time 1 | 21.35 (7.33) | | | |

MBSR: Mindfulness-based stress reduction program, PHQ-9: Patient health questionnaire 9, MAAS: Mindful attention awareness scale, GAD-7: Generalized anxiety disorder scale 7, PSS: perceived stress scale.

in post-test measures, and only the anxiety score significantly changed in the follow-up measures. However, repeated measurements showed better outcomes in anxiety and stress score. In addition, the results provided good insight into the relationship between the number of attended sessions and the improvement in depression and anxiety scores. Furthermore, gender-based stratification revealed that males had significant improvement in mindfulness, anxiety, and stress, while females scored significant change to stress scale.

The literature lacks sufficient information about the effect of MBSR in local non-clinical populations, so these results could provide information on local experience. Unfortunately, the study was interrupted by the COVID-19 pandemic. Although it was not one of our objectives to assess the efficacy of MBSR during the outbreak, this study could also provide information about the effectiveness of the MBSR in highly stressful times.

## Supporting information

**S1 File. The research data.**
(XLSX)

## Acknowledgments

We thank the College of Medicine Research Center (CMRC) at King Saud University for their support and assistance.

## Author Contributions

**Conceptualization:** Ahmed M. Alzahrani, Turky H. Almigbal.

**Data curation:** Ahmed M. Alzahrani, Ahmed Hakami.

**Formal analysis:** Nassr Al-maflehi.

**Funding acquisition:** Ahmad AlHadi, Turky H. Almigbal.

**Methodology:** Ahmed M. Alzahrani, Mohammed H. Aljawadi.

**Project administration:** Turky H. Almigbal.

**Resources:** Ahmad AlHadi, Rawan M. Alotaibi, Muhannad M. Alzahrani, Salem Ali Alammari.

**Supervision:** Mohammed A. Batais, Turky H. Almigbal.

**Writing – original draft:** Ahmed M. Alzahrani, Ahmed Hakami, Rawan M. Alotaibi, Muhannad M. Alzahrani, Salem Ali Alammari.

**Writing – review & editing:** Ahmad AlHadi, Nassr Al-maflehi, Mohammed H. Aljawadi, Mohammed A. Batais, Turky H. Almigbal.

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
