## [Decision Letter · Decision Letter 0]

27 Jul 2023

PONE-D-23-11659The Intricacies of psychological response to mindfulness training among Saudi medical studentsPLOS ONE

Dear Dr. **Alzahrani**,

Thank you for submitting your manuscript to PLOS ONE. After careful consideration, we feel that it has merit but does not fully meet PLOS ONE’s publication criteria as it currently stands. Therefore, we invite you to submit a revised version of the manuscript that addresses the points raised during the review process.

We look forward to receiving your revised manuscript.

Kind regards,

Fadwa Alhalaiqa

Academic Editor

PLOS ONE

Journal Requirements:

Additional Editor Comments:

Dear Authors.

Using of mindfulness to reduce psychological problems among students is very important issues. and this paper is considered the foundation base intervention. However:

1- The title should reflects the aim of the study

2- The abstract must be written in academic way with no more than 250 words that reflect what you have done in your study.

3- The introduction: re-organize it to be in good flow, so the reader could understand what is the gaps in the literatures, importance of your study, why you use your selected instruments.

4- Method: sample size calculation?? based on what intervention group number is different from the control one. randomization process?

- the intervention must be discussed in details. how you have done the recruitment procedure ?? more details about the instruments are needed, validity and reliability? these instruments already translated into Arabic, why you used English version?

Results: is there any differences in demographics characteristics between two groups? since this will may change the statistical tests that you used.

5- Discussion: Usually started by summary of your main study's findings. your point view and rationalization of your findings must be included in the discussion.

6- Limitation: what about the sample size? the dropout rate?

Reviewers' comments:

Reviewer's Responses to Questions

**Comments to the Author**

1. Is the manuscript technically sound, and do the data support the conclusions?

Reviewer #1: Partly

Reviewer #2: Yes

2. Has the statistical analysis been performed appropriately and rigorously? 

Reviewer #1: I Don't Know

Reviewer #2: Yes

3. Have the authors made all data underlying the findings in their manuscript fully available?

Reviewer #1: Yes

Reviewer #2: Yes

4. Is the manuscript presented in an intelligible fashion and written in standard English?

Reviewer #1: Yes

Reviewer #2: Yes

5. Review Comments to the Author

Reviewer #1: Thank you for the opportunity to review this manuscript on a relevant theme. The main limitation is the loss to follow-up of 14 students (10 in the MBSR group and 3 in the waiting list group) on time 1 and 43 students (23 in the MBSR group and 20 in the waiting list group) on time 2.

I hope some questions and comments below contribute to the paper.

#1 - The Abstract should be rewritten in a structured mode (including the following separate sections).

#2 - Abstract - Methods - The authors should indicate the study's design with a commonly used term in the Title or Abstract.

#3 - Abstract - Methods - The study period and sampling and randomization methods should be provided.

#4 - Abstract - Results - It is unclear how many students were evaluated in T1 and T2.

#5 - Abstract - Results - The key results should be present in a numerical form that estimates associations and appropriate measures of variability and uncertainty (e.g., mean difference with 95% confidence intervals, correlation coefficients, and p-values, as appropriate).

#6 - Introduction - The authors should avoid giving the p-values of the previous studies (e.g., the p-values for the results of studies cited as references).

#7 - Methods - The setting should be described in more detail. Besides, the Saudi Arabian medical education system should be explained.

#8 - How was the sample size defined?

#9 - Methods - Lines 148 and 149 - It should be clear that it used stratified random allocation and not simple random allocation.

#10 - Methods -The Perceived Stress Scale (PSS), Mindful Attention Awareness Scale (MAAS), Generalized Anxiety Disorder Scale (GAD-7), and Patient Health Questionnaire 9 (PHQ-9) should be more detailed.

#11 - Methods - The methods should explicitly state all outcomes.

#12 - Methods - What was the normality test used in the study? The authors also should specify the variables that had a parametric distribution.

#13 - Methods - Data Analysis - Lines 198 to 200 - Did the number of attendance at MBSR have a normal distribution? If not, Pearson's correlation coefficient is not an appropriate test to assess this variable.

#14 - Methods - Should repeated measures ANOVA within- and between-subjects design be used to compare MBSR and waiting list groups on T0, T1, and T2?

#15 - Results - The authors should evaluate the balance assignment between the groups regarding the baseline characteristics. In addition, they also should assess whether the loss to follow-up influenced the balance of these variables.

#16 - Results - Legends explaining the acronyms should be included in the tables.

#17 - Discussion - The authors do not need to repeat the data and p-values already given in the Results section.

#18 - Discussion - The low number of students who meet all MBSR sections is noteworthy. To what do the authors attribute this finding? Could the students who noticed the most significant benefit from the initial sessions be the ones who were motivated to do the most sessions?

Reviewer #2: Dear Author,

The manuscript entitled "The Intricacies of psychological response to mindfulness training among Saudi medical students" presents a topic that may interest readers of the journal. Although I consider an interesting topic, I would like to make some comments about the present manuscript below:

introduction:

1. Clarify the context: It would be helpful to provide some context or background information on the alarming findings mentioned at the beginning of the introduction. What specific findings or issues are being addressed? This would help readers understand the significance of the research and why a need-driven response is necessary.

2. Provide more information about mindfulness: While the introduction mentions mindfulness as a solution, it would be beneficial to provide a concise definition or explanation of what mindfulness is. This would enhance the readers' understanding, especially if they are not familiar with the concept.

3. The introduction briefly mentions that the study aims to examine the impact of the Mindfulness-Based Stress Reduction (MBSR) program on medical students, but it would be helpful to explicitly state the research objective. Clearly articulating the purpose of the study would provide a strong foundation and help readers understand the focus of the research.

4. Overall, the introduction provides some relevant information about mindfulness and its potential benefits. However, by addressing the points mentioned above, the authors can improve the clarity and focus of the introduction, making it more engaging for readers.

Methods:

1. Clarify the recruitment process: While the inclusion criteria are mentioned, it would be helpful to provide more details about how the participants were recruited. How were the medical students approached and informed about the study? Providing information on the recruitment process would enhance the transparency of the study and help readers understand how the participants were selected.

2. Provide more information on the waitlist group: The methods mention that there were two study groups, the MBSR group and the waitlist group. However, there is limited information about the waitlist group and their involvement in the study. It would be beneficial to explain the purpose and rationale for including a waitlist group, as well as how they were involved in the study beyond receiving the program at the end.

3. Elaborate on the content of the MBSR program: While the introduction briefly mentions the topics and exercises covered in the MBSR program, it would be valuable to provide more details about each component. This would give readers a clearer understanding of the specific techniques and strategies used in the program.

4. By expanding upon these aspects in the methods section, the authors can provide a more comprehensive explanation of the study design and procedures, enhancing the reader's understanding.

6. PLOS authors have the option to publish the peer review history of their article (what does this mean?). If published, this will include your full peer review and any attached files.

Reviewer #1: No

Reviewer #2: **Yes: **Ebrahim Norouzi

---

## [Author Response · Author response to Decision Letter 0]

10 Oct 2023

Dear reviewers, 

Thank you for your time and effort in reviewing our paper, we are grateful for your valuable input, and it significantly added to our manuscript. 

Editor comment:

Dear Authors,

Using of mindfulness to reduce psychological problems among students is very important issues. and this paper is considered the foundation base intervention. However:

1- The title should reflects the aim of the study

The title has been changed to: The Effectiveness of mindfulness training in improving medical students' stress, depression, and anxiety.

2- The abstract must be written in academic way with no more than 250 words that reflect what you have done in your study.

We have considered all your points and have rewritten the title and abstract in a more organized and clear structure. However, we found it difficult to decrease the word count below 300 without affecting the information needed to represent the research. Also, we included additional information that was requested by the reviewers in the abstract.

3- The introduction: re-organize it to be in good flow, so the reader could understand what is the gaps in the literatures, importance of your study, why you use your selected instruments.

We reviewed the introduction, improved the flow of writing, and emphasized the importance and gap in the literature. The instruments were explained in detail in the methods section, along with their intended use.

4- Method: sample size calculation?? based on what intervention group number is different from the control one. randomization process?

We added the calculation of sample size at line 141. And addressed the difference numbers in groups assignment at line 172, and explained the randomization process at line 169.

- the intervention must be discussed in details. how you have done the recruitment procedure ?? more details about the instruments are needed, validity and reliability? these instruments already translated into Arabic, why you used English version?

The intervention details were added in line 185, and more details about the recruitment and instruments were added in lines 145 and 201, respectively. Although the participants are native Arabic speakers, when it comes to scientific knowledge, it became a norm to use English more than Arabic and they could feel more comfortable using the English version. So, we thought to go with this approach just to make it easy for students. They have a medical background, and it is part of their educational environment to use English. It could be reasonable to use the Arabic version if we are dealing with students who are using Arabic as their scientific language.

Results: is there any differences in demographics characteristics between two groups? since this will may change the statistical tests that you used.

We added the assessment of the differences in baseline characteristics in table 1, we found no significant differences between the study groups.

5- Discussion: Usually started by summary of your main study's findings. your point view and rationalization of your findings must be included in the discussion.

We modified the discussion part so that it starts with the summary of our main findings and rationalization of our findings.

6- Limitation: what about the sample size? the dropout rate?

We added the dropout rate in the limitation section at line 388.

Reviewer #1: 

Thank you for the opportunity to review this manuscript on a relevant theme. The main limitation is the loss to follow-up of 14 students (10 in the MBSR group and 3 in the waiting list group) on time 1 and 43 students (23 in the MBSR group and 20 in the waiting list group) on time 2.

I hope some questions and comments below contribute to the paper.

#1 - The Abstract should be rewritten in a structured mode (including the following separate sections).

We modified the abstract and separated the abstract sections.

#2 - Abstract - Methods - The authors should indicate the study's design with a commonly used term in the Title or Abstract.

We added the study design in the abstract at line 33.

#3 - Abstract - Methods - The study period and sampling and randomization methods should be provided.

We added the study period at line 34, sampling (voluntarily) at line 34, and the randomization method at line 33, due to the required limited number of words in the abstract, we discussed the randomization in more detail in the method section of the paper at line 169.

#4 - Abstract - Results - It is unclear how many students were evaluated in T1 and T2.

We added the number of students in T1 at line 40, and T2 at line 41.

#5 - Abstract - Results - The key results should be present in a numerical form that estimates associations and appropriate measures of variability and uncertainty (e.g., mean difference with 95% confidence intervals, correlation coefficients, and p-values, as appropriate).

We added the measures of variability, mean difference, and 95% CI, correlation coefficients to the result section of the abstract.

#6 - Introduction - The authors should avoid giving the p-values of the previous studies (e.g., the p-values for the results of studies cited as references).

We removed the p values of the previous studies.

#7 - Methods - The setting should be described in more detail. Besides, the Saudi Arabian medical education system should be explained.

We added more information about the setting of the study at line 135.

#8 - How was the sample size defined?

We added the sample size calculation at line 141.

#9 - Methods - Lines 148 and 149 - It should be clear that it used stratified random allocation and not simple random allocation.

We corrected that at line 168.

#10 - Methods -The Perceived Stress Scale (PSS), Mindful Attention Awareness Scale (MAAS), Generalized Anxiety Disorder Scale (GAD-7), and Patient Health Questionnaire 9 (PHQ-9) should be more detailed.

We elaborated more on the instrument starting from line 201.

#11 - Methods - The methods should explicitly state all outcomes.

We clarified all outcomes starting from line 227.

#12 - Methods - What was the normality test used in the study? The authors also should specify the variables that had a parametric distribution.

We added the used normality tests at line 244, And all variables were normally distributed except for the attendance rate to MBSR sessions, it was added at line 267, and we corrected the test of correlation to Spearman’s test. 

#13 - Methods - Data Analysis - Lines 198 to 200 - Did the number of attendance at MBSR have a normal distribution? If not, Pearson's correlation coefficient is not an appropriate test to assess this variable.

It was not normally distributed; we corrected the test of correlation to Spearman’s test.

#14 - Methods - Should repeated measures ANOVA within- and between-subjects design be used to compare MBSR and waiting list groups on T0, T1, and T2?

We added one-way ANOVA to test within and between subjects at line 298.

#15 - Results - The authors should evaluate the balance assignment between the groups regarding the baseline characteristics. In addition, they also should assess whether the loss to follow-up influenced the balance of these variables.

We added the assessment of differences between the groups in Table 1, there were no significant differences, and we added the loss of follow-up in the limitations at line 388.

#16 - Results - Legends explaining the acronyms should be included in the tables.

We added legends explaining the acronyms below the tables.

#17 - Discussion - The authors do not need to repeat the data and p-values already given in the Results section.

We deleted the repeated p-values from the discussion section.

#18 - Discussion - The low number of students who meet all MBSR sections is noteworthy. To what do the authors attribute this finding? Could the students who noticed the most significant benefit from the initial sessions be the ones who were motivated to do the most sessions?

We didn’t assess the reasons for the low number of students who completed the program, and we added that to our limitations at line 390. It could also be attributed to the duration of the intervention and the nature of the intervention, which required them to attend weekly sessions and spend time practicing mindfulness exercises. Which may lead to suboptimal adherence to the program. We recommended that future studies utilize the short version of the Mindfulness program to achieve better adherence.

Reviewer #2: Dear Author,

The manuscript entitled "The Intricacies of psychological response to mindfulness training among Saudi medical students" presents a topic that may interest readers of the journal. Although I consider an interesting topic, I would like to make some comments about the present manuscript below:

introduction:

1. Clarify the context: It would be helpful to provide some context or background information on the alarming findings mentioned at the beginning of the introduction. What specific findings or issues are being addressed? This would help readers understand the significance of the research and why a need-driven response is necessary.

We reorganized the flow of the introduction to be more understandable for the readers and emphasized the purpose of the study and it is importance.

2. Provide more information about mindfulness: While the introduction mentions mindfulness as a solution, it would be beneficial to provide a concise definition or explanation of what mindfulness is. This would enhance the readers' understanding, especially if they are not familiar with the concept.

We elaborated more about that in line 91.

3. The introduction briefly mentions that the study aims to examine the impact of the Mindfulness-Based Stress Reduction (MBSR) program on medical students, but it would be helpful to explicitly state the research objective. Clearly articulating the purpose of the study would provide a strong foundation and help readers understand the focus of the research.

We mentioned the objectives in a clearer manner at line 122.

4. Overall, the introduction provides some relevant information about mindfulness and its potential benefits. However, by addressing the points mentioned above, the authors can improve the clarity and focus of the introduction, making it more engaging for readers.

We addressed all the suggestions and comments and made the required modifications accordingly.

Methods:

1. Clarify the recruitment process: While the inclusion criteria are mentioned, it would be helpful to provide more details about how the participants were recruited. How were the medical students approached and informed about the study? Providing information on the recruitment process would enhance the transparency of the study and help readers understand how the participants were selected.

We added more details about the recruitment process and how we invited the participants starting from line 144.

2. Provide more information on the waitlist group: The methods mention that there were two study groups, the MBSR group and the waitlist group. However, there is limited information about the waitlist group and their involvement in the study. It would be beneficial to explain the purpose and rationale for including a waitlist group, as well as how they were involved in the study beyond receiving the program at the end.

We added more information about the use of the control group at line 181, and there were no other roles or tasks required from them, other than completing the measures at the same time intervals. 

3. Elaborate on the content of the MBSR program: While the introduction briefly mentions the topics and exercises covered in the MBSR program, it would be valuable to provide more details about each component. This would give readers a clearer understanding of the specific techniques and strategies used in the program.

We added more details of the program's eight sessions starting from line 186.

4. By expanding upon these aspects in the methods section, the authors can provide a more comprehensive explanation of the study design and procedures, enhancing the reader's understanding.

We added more details about the study design, recruitment, randomization, instruments, and outcomes in the method section of the manuscript, so it would make the procedure clearer for the reader.

Thank you,

On behalf of the research team, 

Ahmed Alzahrani,

---

## [Decision Letter · Decision Letter 1]

16 Oct 2023

The Effectiveness of mindfulness training in improving medical students' stress, depression, and anxiety

PONE-D-23-11659R1

Dear Dr. Ahmed Alzahrani,

We’re pleased to inform you that your manuscript has been judged scientifically suitable for publication and will be formally accepted for publication once it meets all outstanding technical requirements.

Kind regards,

Fadwa Alhalaiqa

Academic Editor

PLOS ONE

Additional Editor Comments (optional):

Reviewers' comments:

Reviewer's Responses to Questions

**Comments to the Author**

1. If the authors have adequately addressed your comments raised in a previous round of review and you feel that this manuscript is now acceptable for publication, you may indicate that here to bypass the “Comments to the Author” section, enter your conflict of interest statement in the “Confidential to Editor” section, and submit your "Accept" recommendation.

Reviewer #1: All comments have been addressed

Reviewer #2: All comments have been addressed

2. Is the manuscript technically sound, and do the data support the conclusions?

Reviewer #1: Yes

Reviewer #2: Yes

3. Has the statistical analysis been performed appropriately and rigorously? 

Reviewer #1: Yes

Reviewer #2: Yes

4. Have the authors made all data underlying the findings in their manuscript fully available?

Reviewer #1: Yes

Reviewer #2: Yes

5. Is the manuscript presented in an intelligible fashion and written in standard English?

Reviewer #1: Yes

Reviewer #2: Yes

6. Review Comments to the Author

Reviewer #1: The authors have made appropriate adjustments to the original submission.

All my comments have been answered, and I have no further recommendations.

Reviewer #2: The authors have addressed all of my concerns. I would like to accept this manuscript as it is in the present version. I really like this manuscript.

7. PLOS authors have the option to publish the peer review history of their article (what does this mean?). If published, this will include your full peer review and any attached files.

Reviewer #1: **Yes: **Fábio Ferreira Amorim

Reviewer #2: **Yes: **Ebrahim Norouzi

---

## [Editor Report · Acceptance letter]

20 Oct 2023

PONE-D-23-11659R1 

The Effectiveness of mindfulness training in improving medical students' stress, depression, and anxiety 

Dear Dr. Alzahrani:

I'm pleased to inform you that your manuscript has been deemed suitable for publication in PLOS ONE. Congratulations! Your manuscript is now with our production department. 

Kind regards, 

on behalf of

Pro Fadwa Alhalaiqa 

Academic Editor

PLOS ONE